# Knowledge on Infection Prevention and Control and associated factors among undergraduate health professional students at Makerere University College of Health Sciences, Uganda

**Racheal Nalunkuma**[1], **Jonathan Nkalubo**[1,2]*, **Derrick Bary Abila**[1,3]

**1** Makerere University College of Health Sciences, Kampala, Uganda, **2** Mulago National Referral Hospital, Kampala, Uganda, **3** Faculty of Biology, Medicine, and Health, University of Manchester, Manchester, United Kingdom

* nkalubo11@gmail.com

## Abstract

### Introduction

To practice adequate Infection Prevention and Control (IPC) measures, health professional students need to have adequate knowledge of IPC. In this study, we assessed the knowledge of health professional students at Makerere University College of Health Sciences on Infection Prevention and Control.

### Methods

We conducted a cross-sectional online survey among health professional students studying at Makerere University College of Health Sciences located in Kampala, Uganda. An adapted questionnaire was used to measure knowledge on Infection Prevention and Control among students.

### Results

A total of 202 health professional students were included in the study. The mean age was 24.43 years. Majority were male 63.37% (n = 128), from the school of medicine 70.79% (n = 143) and used one source of information for IPC 49.50% (n = 100). Being in year three (Adjusted coefficient, 6.08; 95% CI, 2.04–10.13; p-value = 0.003), year four (Adjusted coefficient, 10.87; 95% CI, 6.91–14.84; p < 0.001) and year five (Adjusted coefficient, 8.61; 95% CI, 4.45–12.78; p < 0.001) were associated with a higher mean in total percentage score of knowledge on IPC compared to being in year one.

### Conclusion

IPC knowledge was good among health professional students in Makerere University although more emphasis is needed to improve on their IPC knowledge in various sections

**Data Availability Statement:** The data underlying the results presented in the study are available from https://figshare.com/articles/dataset/IPC_

Study_dataset_coded_Jan_03_2021_final_used_inanalysis_csv/14621247 https://doi.org/10.6084/m9.figshare.14621247.v1.

**Funding:** The author(s) received no specific funding for this work.

**Competing interests:** The authors have declared that no competing interests exist.

like hand hygiene. Infection Prevention and Control courses can be taught to these students starting from their first year of university education.

## Introduction

Globally, infections remain the biggest burden in health care service delivery, causing a major setback due to increased health care costs in the long [1]. According to the International Federation of Infection Control (2007), Infection Prevention and Control (IPC) has been regarded as a vital substructure of the health care system and thus the need to adhere to the standard protocols to prevent and lessen the risk of infectious disease transmission at the health facilities among patients, staff, and visitors [2, 3]. Infection prevention and control can be defined as "policies, procedures, and activities which aim to prevent or minimize the risk of transmission of infectious diseases at healthcare facilities" [4].

Prioritization of Health Care Worker's (HCW's) safety calls for a demonstration of practical and evidence-based methods of high IPC standards to protect them from Health Worker Associated Infections (HCAIs), decreasing the adverse socioeconomic and psychological impact. Thus strong programs and policies are a cornerstone to a resilient health system effective in prevention, detection, and response to public health crises and disease outbreaks [4, 5].

Having declared Coronavirus-2019 (COVID-19) on 12 March 2020 as a global pandemic and defined as "a respiratory disease caused by SARS-CoV-2 that emerged in China in 2019" [3], with no standard treatment for the disease but only supportive care, HCW are at an increased risk of contracting this deadly infection as they are the main frontiers in the management of this disease. Thousands of Health Care Workers (HCWs) have been reported to have been infected in the process of offering medical services to the COVID-19 patients and some have lost their lives due to this deadly disease [6, 7].

As the Ugandan government fights the increasing COVID-19 cases and starts opening up learning institutions, it is important to assess the IPC knowledge among health professional students as they are at a high risk of contracting the disease. With the various IPC methods proposed, health professional students are expected to have adequate knowledge about them to reduce the risk of infection transmission between them and the patients while learning and offering health care services and also reduce healthcare-associated infections [8]. Studies on knowledge on IPC measures can identify IPC-related knowledge gaps and generate evidence to support and provide the necessary training to health professional students. The study aimed to assess the knowledge of health professional students at Makerere University College of Health Sciences on Infection Prevention and Control.

## Methods

### Study design

This was an online descriptive cross-sectional study design that involved the use of quantitative methods. The study data collection was conducted from October 2020 to December 2020.

### Study area

The study was conducted at Makerere University, College of Health Sciences located on Upper Mulago Hill in Kampala, the Capital of Uganda. The College of Health Sciences has four schools which include the School of Biomedical Sciences, School of Health Sciences, School of

Medicine, and School of Public Health. These schools comprise health professional students. The duration of courses/programs ranges from three to five years.

## Target population

The study included consented students studying at Makerere University College of Health Sciences irrespective of the year of study. A minimum sample size of 138 medical students was reached using the Kish and Leslie (1965) formula for cross-sectional studies. The following assumptions were made; (1) p = 0.2, assuming that the attitude and knowledge of infection prevention and control are not favorable among 10% of medical students in Uganda; (2) Z score of 1.96 corresponding with 95% confidence interval; and (3) d value of 0.05, which was the maximum error. The formula is displayed below.

$$N = \frac{p(1-p)Z^2}{d^2}$$

We recruited 207 participants in 3 months (October, November, and December 2020) and excluded five students from the analysis since they had not provided consent.

## Sampling procedure

We used convenience sampling where a link to an online Google form was shared within class WhatsApp groups and email lists of health professional students at Makerere University College of Health Sciences.

## Data collection tool

We used a modified questionnaire from a study that assessed knowledge on Infection Prevention and Control among students [9].

The independent variables studied included demographic characteristics like sex, age in years, year of study, and school of attachment. The dependent variables measured knowledge on IPC. These comprised of aspects of IPC guidelines like hand hygiene, knowledge about isolation precautions, respiratory hygiene, and cough etiquette, and the use of personal protective equipment.

## Data collection methods

Data was collected through a self-administered online and anonymous questionnaire consisting of 42 questions that included sections on demographic characteristics and IPC knowledge. A link to the survey was sent to students via email, text message, and through class WhatsApp groups. An information page and consent forms were included in the first part of the questionnaire. Only participants who consent to participate in the study by clicking the yes option will continue to the questionnaire.

## Data management

For all sections of the questionnaire, categorical variables were coded to numerical values to allow for measures of association and regression analysis to be performed. For a statement or question, a correct response was scored 1 and an incorrect response was scored 0. The score sheet for all the statements/questions is attached (**S1 Appendix**). The total score for each section was calculated and converted into a percentage score. Also, the total score (maximum of 40) for all the questions/statements was calculated.

## Data analysis

The demographic variables were summarized as descriptive statistics i.e., means, medians, and proportions. Using Bivariate and multivariate linear regression analysis, we tested the association between demographic characteristics and the total percentage of correct answers for all questions. In Bivariate analysis, the independent variables considered were age, sex, year of study, school of study, and the number of sources of information on IPC. The dependent/outcome variable was the percentage score of correct answers for all questions. In the multivariate analysis, the independent variables considered were age, sex, year of study, school of study, and the number of sources of information on IPC. For both the bivariate and multivariate analysis, we considered a 95% confidence interval and a significance level of less than 0.05.

## Quality control

Questions were designed in simple English words for effective comprehension by the medical students. Soft copies of the questionnaires were designed with checks to allow valid and complete entry only. All submitted forms were checked for completeness and all those missing more than 2 items were to be discarded but all were complete at end of data collection. The questionnaire was pretested among 10 students to assess their validity, reliability, and bias. Additionally, the principal investigator directly supervised all the activities from the beginning to the completion of the study.

## Ethical considerations

Research ethical approval was granted by Mulago Hospital Research and Ethics Committee. Participation in the study was entirely voluntary and online written consent was obtained by checking a box on the online form to signifying that informed consent was granted. The participants' identifiers were not captured by the online questionnaire. Data generated from the study was only used for research purposes.

## Results

### Demographic characteristics

Of the 202 students included in the study analysis, the mean age was 24.43 years with a standard deviation of 4.22. The median age was 24 years with an interquartile range of 19–38 years. Most of the participants were male (63.37%, 128/202), in the fourth and fifth year of their study (25.25%, 51/202 and 22.77%, 46/202 respectively), and were from the school of medicine (70.79%, 143/202). Most of the students reported being self-learned about Infection Prevention and Control (75.74%, 153/202) with most of them using only one source of information (49.50%, 100/202) **Table 1**.

### Knowledge about various aspects of infection prevention and control

In the section on the general concept of Infection Prevention and Control (IPC), the least correctly answered statement was "all body fluids except sweat should be viewed as sources of infection" with 31.68% (64 out of 202 responses) of the responses correct. In the section on hand hygiene, no student got the response of "In standard handwashing: the minimum duration should be 30 seconds" correct. In the section on personal protective equipment (PPEs), the least correctly answered was, "gloves should be changed between different procedures on the same patient," with 21.29% (43 out of 202 responses) of the responses correct. In the section on sharps disposal and sharp injuries, the least correctly answered statement was, "soiled

**Table 1. Demographic characteristics of health professional students at Makerere University College of Health Sciences, Uganda.**

| | Frequency (n) | Percentage (%) |
|---|---|---|
| **Age** | | |
| Mean (Standard deviation) | 24.43 (4.22) | |
| Median (Inter Quartile Range) | 24 (19–38) | |
| **Sex** | | |
| Female | 74 | 36.63% |
| Male | 128 | 63.37% |
| **Year of study** | | |
| Year One | 35 | 17.33% |
| Year Two | 29 | 14.36% |
| Year Three | 41 | 20.30% |
| Year Four | 51 | 25.25% |
| Year Five | 46 | 22.77% |
| **School of study** | | |
| School of Medicine | 143 | 70.79% |
| School of Health Sciences | 31 | 15.35% |
| School of Biomedical Sciences | 18 | 8.91% |
| School of Public Health | 10 | 4.95% |
| **Source of information on Infection Prevention and Control (IPC)** | | |
| Self-Learning | 153 | 75.74% |
| Informal practical learning onwards | 17 | 8.42% |
| Formal curricular teaching | 19 | 9.41% |
| Infection control courses | 10 | 4.95% |
| Internet | 2 | 0.99% |
| Media | 1 | 0.50% |
| **Number of sources of information** | | |
| 1 | 100 | 49.50% |
| 2 | 52 | 25.74% |
| 3 | 33 | 16.34% |
| 4 | 16 | 7.92% |
| 5 | 1 | 0.50% |

sharps objects should be shredded (cut into tiny pieces) before final disposal," with 17.82% (36 out of 200) of the responses correct (**Table 2**).

In the section on respiratory hygiene, the least correctly answered statement was, "Cough/ sneeze on a disposable napkin and wash your hands," with 51.98% (105 out of 202) of the responses correct. In the section on care for healthcare providers, the least correctly answered statement was, "post-exposure immunization prevents the risk of hepatitis B infection following exposure," with 36.63% (74 out of 202) of the responses correct (**Table 2**).

## Factors associated with the total percentage score of correct answers for all questions responded to by health professional students

Bivariate and multivariate linear regression analysis was used to assess the association between the student demographic factors and the total percentage score of correct answers for all questions responded to. Increase in age (Adjusted Coefficient 0.62; 95% Confidence Interval CI, 0.32–0.92; P-value< 0.001) was associated with a higher mean in total percentage score. Also,

**Table 2. Proportion of correct responses in knowledge on various aspects of infection prevention and control.**

| | Frequency of Correct Responses (n = 202) | The proportion of Correct Responses |
|---|---|---|
| **Section A: General concept of Infection Prevention and Control (IPC)** | | |
| The main goal of infection control is? (1 option) | 195 | 96.53% |
| Definition of standard precautions? (1 option) | 180 | 89.11% |
| All patients are sources of infections regardless of their diagnoses (true) | 165 | 81.68% |
| All body fluids except sweat should be viewed as sources of infection (true) | 64 | 31.68% |
| **Total score General concept of IPC** | | |
| Mean (Standard deviation) Percentage score | 74.75% (17.63%) | |
| Median Percentage score (Interquartile range) | 75% (25%– 100%) | |
| **Section B: Hand Hygiene** | | |
| Hand washing minimizes microorganisms acquired on the hands if hands are soiled (true) | 182 | 90.10% |
| Handwashing reduces the incidence of healthcare-related infections (true) | 194 | 96.04% |
| In standard handwashing: the minimum duration should be. . . (1 option) | 0 | 0% |
| Hand decontamination: includes washing the. . . .. . . .. with antiseptic soap for 30 seconds (1 option) | 29 | 14.36% |
| Alcohol hand rub substitutes hand washing even if the hands are soiled (false) | 145 | 71.78% |
| Hand washing is indicated between tasks and procedures on the same patient (true) | 122 | 60.40% |
| The use of gloves replaces the need for handwashing (false) | 180 | 89.11% |
| Hand washing is indicated after removal of gloves (true) | 182 | 90.10% |
| Hand washing is needed with patients with respiratory infections including COVID 19 (true) | 197 | 97.52% |
| **Total Score for Hand Hygiene** | | |
| Mean (Standard deviation) Percentage score | 67.71% (12.57%) | |
| Median Percentage score (Interquartile range) | 66.67% (33.33%– 88.89%) | |
| **Section C: Personal Protective Equipment (PPE)** | | |
| PPEs such as masks and head caps provide protective barriers against infection (true) | 197 | 97.52% |
| Use of PPEs eliminate the risk of acquiring occupational infections (true) | 166 | 82.18% |
| PPEs are exclusively suitable to laboratory and cleaning staff for their protection (false) | 94 | 46.53% |
| PPEs should be used only whenever there is contact with blood (false) | 186 | 92.08% |
| Gloves and masks can be re-used after proper cleaning (false) | 163 | 80.69% |
| Used PPEs are to be discarded through regular dust bins (false) | 134 | 66.34% |
| Gloves should be changed between different procedures on the same patient (true) | 43 | 21.29% |
| Masks made of cotton or gauze are most protective (false) | 101 | 50.00% |
| Masks and gloves can be re-used if dealing with same patient (false) | 155 | 76.73% |
| **Total Score for PPE** | | |
| Mean (Standard deviation) Percentage score | 68.15% (16.31%) | |

*(Continued)*

**Table 2.** (Continued)

| | Frequency of Correct Responses (n = 202) | The proportion of Correct Responses |
|---|---|---|
| Median Percentage score (Interquartile range) | 66.67% (33.33%– 88.89%) | |
| **Section D: Sharps disposal and Sharp Injuries** | | |
| Used needles should be recapped after use to prevent injuries (false) | 80 | 39.60% |
| Used needles should be bent after use to prevent injuries (false) | 156 | 77.23% |
| Sharps container is labelled with…(1 option) | 117 | 59.09% |
| Soiled sharps objects should be shredded (cut into tiny pieces) before final disposal (true) | 36 | 17.82% |
| Sharps injuries should be managed with no need of reporting (false) | 184 | 91.09% |
| Needle-stick injuries are the least commonly encountered in general practice (false) | 144 | 71.29% |
| Post-exposure prophylaxis is used for managing Needle-stick injuries from an HIV-infected patient (true) | 172 | 85.15% |
| Immediate management of sharps injuries includes… (1 option) | 103 | 50.99% |
| **Total Score for Sharps disposal and Sharp Injuries** | | |
| Mean (median) Percentage score | 61.55% (20.50%) | |
| Median Percentage score (Interquartile range) | 62.50% (12.5%– 87.50%) | |
| **Section E: Respiratory hygiene and cough etiquette** | | |
| Cough/sneeze on a disposable napkin and wash your hands (True) | 190 | 94.06% |
| Cough/sneeze over the shoulder if a napkin is not available (True) | 105 | 51.98% |
| Keep a distance of 3 feet from others when coughing (true) | 176 | 87.13% |
| Wipe your hands on the inside of your white coat after you cough or sneeze (false) | 177 | 87.62% |
| **Total Score for Respiratory hygiene and cough etiquette** | | |
| Mean (Standard deviation) Percentage score | 80.2% (19.42%) | |
| Median Percentage score (Interquartile range) | 75% (25%– 100%) | |
| **Section F: Care of Healthcare Providers** | | |
| Immunization history of health care providers should be obtained before recruitment (true) | 178 | 88.12% |
| The risk for a health provider to acquire HIV infection after a needle-stick injury is… (option) | 55 | 27.23% |
| Post-exposure immunization prevents the risk of hepatitis B infection following exposure (true) | 74 | 36.63% |
| For the prevention of hepatitis B, immunizations are recommended for all healthcare workers (true) | 193 | 95.54% |
| Following exposure to a patient with flu, antibiotics are required for the prevention of infection (false) | 144 | 71.29% |
| Health providers with the highest risk of exposure to tuberculosis include radiologists (true) | 97 | 48.02% |
| **Total Score for Care of Healthcare Providers** | | |
| Mean (Standard deviation) Percentage score | 61.14% (17.3%) | |
| Median Percentage score (Interquartile range) | 66.67% (33.33%– 100%) | |
| **Average for all total scores for each parameter (n = 198)** | | |
| Mean (Standard deviation) Percentage score | 67.51% (10.16%) | |
| Median Percentage score (Interquartile range) | 70% (42.5%– 82.5%) | |

**Table 3. Factors associated with the total percentage score of correct answers for all questions responded to by health professional students at Makerere University College of Health Sciences, Uganda.**

| | Observations | Mean of Total Percentage Score (SD) | Bivariate Analysis | | Multivariate Analysis | |
|---|---|---|---|---|---|---|
| | | | & Crude Coefficient (95% CI) | P—value | & Adjusted Coefficient (95% CI) | P—value |
| **Age** | | | | | | |
| A one-year increase in age | 198 | - | 0.81 (0.49–1.12) | < 0.001 | 0.62 (0.32–0.92) | **< 0.001** |
| **Sex** | | | | | | |
| Female | 73 | 67.33 (9.49) | **Reference** | | **Reference** | |
| Male | 125 | 67.62 (10.57) | 0.29 (-2.67–3.25) | | -0.05 (-2.61–2.51) | 0.970 |
| **Year of study** | | | | | | |
| Year One | 35 | 60.21 (8.77) | **Reference** | | **Reference** | |
| Year Two | 28 | 60.54 (11.35) | 0.32 (-4.11–4.75) | 0.886 | -0.93 (-5.28–3.43) | 0.675 |
| Year Three | 39 | 68.08 (8.12) | 7.86 (3.8–11.93) | < 0.001 | 6.08 (2.04–10.13) | **0.003** |
| Year Four | 51 | 72.74 (7.35) | 12.53 (8.7–16.36) | < 0.001 | 10.87 (6.91–14.84) | **< 0.001** |
| Year Five | 45 | 71.11 (9.31) | 10.9 (6.96–14.83) | < 0.001 | 8.61 (4.45–12.78) | **< 0.001** |
| **School of study** | | | | | | |
| School of Medicine | 139 | 68.29 (9.81) | **Reference** | | **Reference** | |
| School of Health Sciences | 31 | 65.64 (11.22) | -2.65 (-6.63–1.34) | 0.192 | 0.77 (-2.73–4.28) | 0.664 |
| School of Biomedical Sciences | 18 | 65.28 (10.94) | -3.01 (-8.04–2.01) | 0.238 | 3.89 (-0.71–8.48) | 0.097 |
| School of Public Health | 10 | 66.5 (10.29) | -1.79 (-8.36–4.77) | 0.591 | 1.08 (-4.82–6.98) | 0.718 |
| **Number of sources of information** | | | | | | |
| One | 98 | 65.87 (10.66) | **Reference** | | **Reference** | |
| Two | 50 | 66.65 (8.83) | 0.78 (-2.62–4.18) | 0.650 | -0.84 (-3.86–2.17) | 0.581 |
| Three | 33 | 70.15 (9.84) | 4.28 (0.34–8.22) | 0.033 | 2.54 (-0.93–6) | 0.150 |
| Four | 16 | 74.22 (8.40) | 8.35 (3.08–13.63) | 0.002 | 6.27 (1.59–10.95) | **0.009** |
| Five (One observation) | 1 | 77.5 (-) | 11.63 (-8.03–31.3) | 0.245 | 8.76 (-8.24–25.76) | 0.311 |

& The coefficient is the mean difference in total scores. It was calculated using the simple and multiple linear regression models reporting the crude and adjusted coefficients, respectively.

being in year three (Adjusted coefficient, 6.08; 95% CI, 2.04–10.13; p-value = 0.003), year four (Adjusted coefficient, 10.87; 95% CI, 6.91–14.84; p < 0.001), and year five (Adjusted coefficient, 8.61; 95% CI, 4.45–12.78; p < 0.001) were associated with a higher mean in total percentage score of knowledge on IPC compared to being in year one. Students who used four sources (Adjusted Coefficient, 6.27; 95% CI, 1.59–10.95; p-value = 0.009) of information for gaining knowledge of IPC had a higher mean score in the knowledge of IPC compared to those who used only one source (**Table 3**).

## Discussion

Occupational acquired infection are the leading cause of morbidity and mortality among health care workers and the common HCAIs include; Human Immune Virus and Acquired Immune Deficiency Syndrome (HIV/AIDS), Tuberculosis, Hepatitis B, and bacterial infections [10] and currently COVID-19 [11]. In this study, we aimed to assess the knowledge of health professional students at Makerere University College of Health Sciences, Uganda on Infection Prevention and Control. We used an online questionnaire powered by Google Forms to collect the data among the students. From this study, we found that students in year three, four and year students had significantly higher knowledge on the combined aspects of

Infection Prevention and Control when compared to those in year one. Although the students had a good knowledge on Infection Prevention and control, low level of knowledge was observed in sections on disposal of sharps and care for healthcare professionals.

All the 202 participants demonstrated fair knowledge of hand hygiene though they all lacked knowledge about the standard hand washing time. In contrast, a study carried out by [12] showed that 79% knew the correct 30 seconds duration of hand hygiene. This implies that failure to have sufficient knowledge about the duration of handwashing increases the risk of infection transmission between patients by health care workers.

Of the 202 respondents, the majority (n = 64, 31.68%) reported all body fluids except sweat to be viewed as a source of infection which was consistent with a study done in Saudi Arabia [9]. This shows that students though in different countries have inadequate knowledge about body fluids being one of the modes of infection spread between health care workers and patients. A total of 43 (21.29%) of the respondents knew the benefits of changing gloves between different procedures on the same patient. This is inconsistent with a study carried out in Saudi Arabia where 54.3% (n = 70) knew its benefits [9]. The use of a single pair of a glove on different body sites on a patient increases the risk of transfer of microorganisms thus spread of infections.

The section on disposal of sharps and care for healthcare professionals has the least level of knowledge among the students in this study. This consistent with findings from a similar study among health professional students in Saudi Arabia, where these sections also scored low in terms of the level of knowledge on disposal of sharps and care for healthcare workers [9]. This could be because there has been an emphasis on respiratory hygiene during the COVID-19 pandemic period when the study was conducted.

Also, from this study, students in years three, four, and five had higher percentage scores on knowledge from all questions compared to those in year one. These findings are consistent with findings from a study among students in Peru where it was reported that clinical year students had better knowledge compared to pre-clinical students [13]. This could be because the first-year students have not yet been introduced to the various aspects of infection prevention and control while the students in years three, four, and five have been exposed to these concepts since their training is more practical (clinical).

## Conclusions

Infection prevention and control (IPC) knowledge was good among health care professionals at the College of Health Sciences, Makerere University. However, more emphasis is needed to improve on the students' IPC knowledge in various sections like hand hygiene.

## Recommendations

Infection Prevention and Control courses should be taught to students of health care professionals starting from their first year in medical school since our study showed poor IPC knowledge among non-clinical students in years one, two, and three.

## Supporting information

**S1 Appendix. Questionnaire.**
(DOCX)

## Acknowledgments

The authors would like to thank the students at Makerere University College of Health Sciences for agreeing to participate in this study. We extend our sincere appreciation to Mr. Abdullah Mohammed and his colleagues for allowing us to adapt their study tool.

## Author Contributions

**Conceptualization:** Racheal Nalunkuma, Jonathan Nkalubo.

**Data curation:** Racheal Nalunkuma, Jonathan Nkalubo, Derrick Bary Abila.

**Formal analysis:** Racheal Nalunkuma, Jonathan Nkalubo, Derrick Bary Abila.

**Investigation:** Racheal Nalunkuma, Jonathan Nkalubo.

**Methodology:** Jonathan Nkalubo.

**Project administration:** Racheal Nalunkuma, Jonathan Nkalubo.

**Resources:** Jonathan Nkalubo.

**Software:** Jonathan Nkalubo, Derrick Bary Abila.

**Supervision:** Racheal Nalunkuma, Jonathan Nkalubo.

**Validation:** Racheal Nalunkuma, Jonathan Nkalubo.

**Visualization:** Racheal Nalunkuma, Jonathan Nkalubo.

**Writing – original draft:** Jonathan Nkalubo.

**Writing – review & editing:** Racheal Nalunkuma, Jonathan Nkalubo, Derrick Bary Abila.

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
