## [Decision Letter · Decision Letter 0]

4 May 2021

PONE-D-21-09130

Knowledge on Infection Prevention and Control and Associated Factors among Undergraduate Health Professional Students at Makerere University College of Health Sciences, Uganda

PLOS ONE

Dear Dr. Nkalubo,

Thank you for submitting your manuscript to PLOS ONE. After careful consideration, we feel that it has merit but does not fully meet PLOS ONE’s publication criteria as it currently stands. Therefore, we invite you to submit a revised version of the manuscript that addresses the points raised during the review process.

We look forward to receiving your revised manuscript.

Kind regards,

Prasenjit Mitra, MD, MRSB, MIScT, FLS, FACSc, FAACC

Academic Editor

PLOS ONE

Journal Requirements:

Please make the corrections as suggested by the reviewer.

Reviewers' comments:

Reviewer's Responses to Questions

**Comments to the Author**

1. Is the manuscript technically sound, and do the data support the conclusions?

Reviewer #1: Yes

2. Has the statistical analysis been performed appropriately and rigorously? 

Reviewer #1: Yes

3. Have the authors made all data underlying the findings in their manuscript fully available?

Reviewer #1: Yes

4. Is the manuscript presented in an intelligible fashion and written in standard English?

Reviewer #1: Yes

5. Review Comments to the Author

Reviewer #1: Introduction

Reword the sentence on

Paragraph 3, Line 69-70: ….” through which schools can be opened, this questions the IPC knowledge of medical students as their at a higher risk of contracting this disease”

Data Managent

Consider deleting statement on Line 115 – 116, it has been repeat on Line 117-118

Discussion

Clearly define clinical year and non-clinical year students

Check the sentence and correct appropriately

Line 222-223: “…… in equal response of 64 (49.6%) in a study carried out be (Khubrani et al., 2018)……..

Please check in-text citation on Line 227…….. It should read “In contrast to a study carried out by Khubrani et al., (2018) NOT In contrast to a study carried out by (Khubrani et al., 2018)”

Conclusion

On Line 245, “knowledge was good among health care professions at the College of Health….”

Professions should be changed to professionals

6. PLOS authors have the option to publish the peer review history of their article (what does this mean?). If published, this will include your full peer review and any attached files.

Reviewer #1: No

---

## [Author Response · Author response to Decision Letter 0]

20 May 2021

Nkalubo Jonathan

Makerere University

College of Health Sciences

Upper Mulago Hill Road

Kampala, Uganda

May 11, 2021

To Prasenjit Mitra,

Academic Editor

PLOS ONE

RE: RESPONSE TO EDITOR AND REVIEWER COMMENTS

Kindly find below the responses to the editor and reviewer comments.

Response: Thank you for coordinating the review process. We have revised the manuscript and addressed the Reviewer’s comments.

Response: We have attached a rebuttal letter, a marked-up copy of our manuscript, and an unmarked version of our paper without tracked changes

5. Review Comments to the Author

Reviewer #1: Introduction

Reword the sentence on

Paragraph 3, Line 69-70: ….” through which schools can be opened, this questions the IPC knowledge of medical students as their at a higher risk of contracting this disease.”

Response: We have reworded the sentence.

Data Management

Consider deleting statement on Line 115 – 116, it has been repeat on Line 117-118

Response: We have deleted Line 115 – 116.

Discussion

Reviewer comment: Clearly define clinical year and non-clinical year students.

Response: We have rephrased the statement in the discussion to compare students in years three, four and five with students in year one.

Reviewer comment: Check the sentence and correct appropriately

Line 222-223: “…… in equal response of 64 (49.6%) in a study carried out be (Khubrani et al., 2018)……..

Response: The statement has been rephrased.

Reviewer comment: Please check in-text citation on Line 227…….. It should read “In contrast to a study carried out by Khubrani et al., (2018) NOT In contrast to a study carried out by (Khubrani et al., 2018)”

Response: The statement has been rephrased.

Reviewer comment: Conclusion

On Line 245, “knowledge was good among health care professions at the College of Health….”

Professions should be changed to professionals.

Response: The grammatical error has been corrected.

On behalf of my co-authors, I thank you for your consideration of this resubmission. We appreciate your time and look forward to your response.

Sincerely,

Nkalubo Jonathan, MD

---

## [Decision Letter · Decision Letter 1]

14 Jun 2021

PONE-D-21-09130R1

Knowledge on Infection Prevention and Control and Associated Factors among Undergraduate Health Professional Students at Makerere University College of Health Sciences, Uganda

PLOS ONE

Dear Dr. Nkalubo,

Thank you for submitting your manuscript to PLOS ONE. After careful consideration, we feel that it has merit but does not fully meet PLOS ONE’s publication criteria as it currently stands. Therefore, we invite you to submit a revised version of the manuscript that addresses the points raised during the review process.

ACADEMIC EDITOR: Please revise the manuscript according to reviewer's instruction

We look forward to receiving your revised manuscript.

Kind regards,

Prasenjit Mitra, MD, MRSB, MIScT, FLS, FACSc, FAACC

Academic Editor

PLOS ONE

Journal Requirements:

Reviewers' comments:

Reviewer's Responses to Questions

**Comments to the Author**

1. If the authors have adequately addressed your comments raised in a previous round of review and you feel that this manuscript is now acceptable for publication, you may indicate that here to bypass the “Comments to the Author” section, enter your conflict of interest statement in the “Confidential to Editor” section, and submit your "Accept" recommendation.

Reviewer #1: All comments have been addressed

2. Is the manuscript technically sound, and do the data support the conclusions?

Reviewer #1: (No Response)

3. Has the statistical analysis been performed appropriately and rigorously? 

Reviewer #1: Yes

4. Have the authors made all data underlying the findings in their manuscript fully available?

Reviewer #1: Yes

5. Is the manuscript presented in an intelligible fashion and written in standard English?

Reviewer #1: Yes

6. Review Comments to the Author

Reviewer #1: The manuscript is well written and all comments have been addressed. A few typographical errors have been identified and have been marked in the attached document.

METHODS

Please elaborate how how the sample size for the study was determined.

RESULTS

Also please remove all notes from the tables. Convert them to footnotes for each table

DISCUSION"

Line 264: ".....have had any exposure to these concepts....." should rather read "......have been exposed to these concepts..."

CONLCUSION

Line 270: "..........at the College of Health Sciences, Makerere University though......." should read ".......at the College of Health Sciences, Makerere University. However, .........."

7. PLOS authors have the option to publish the peer review history of their article (what does this mean?). If published, this will include your full peer review and any attached files.

Reviewer #1: No

---

## [Author Response · Author response to Decision Letter 1]

18 Jun 2021

Nkalubo Jonathan

Makerere University

College of Health Sciences

Upper Mulago Hill Road

Kampala, Uganda

June 18, 2021

To

Prasenjit Mitra,

Academic Editor

PLOS ONE

RE: Response to reviewer comments:

Kindly find below the response to the reviewer comments.

Reviewer #1: 

The manuscript is well written and all comments have been addressed. A few typographical errors have been identified and have been marked in the attached document.

Response: The typographical errors have been rectified.

METHODS

Reviewer comment: Please elaborate how the sample size for the study was determined.

Response: A statement explaining how the sample size was reached has been included under the section of “target population.”

RESULTS

Reviewer comment: Also, please remove all notes from the tables. Convert them to footnotes for each table.

Response: The foot notes have been removed from the tables and placed below the respective tables.

DISCUSION

Reviewer comment: Line 264: ".....have had any exposure to these concepts....." should rather read "......have been exposed to these concepts..."

Response: The statement has been rephrased.

CONLCUSION

Reviewer comment: Line 270: "..........at the College of Health Sciences, Makerere University though......." should read ".......at the College of Health Sciences, Makerere University. However, .........."

Response: The statement has been rephrased.

Furthermore, we have included a section on data availability at the end of the methods and materials section. We have also revised all references to meet the journal requirement.

On behalf of my co-authors, I thank you for your consideration of this resubmission. We appreciate your time and look forward to your response.

Sincerely,

Nkalubo Jonathan, MD

---

## [Decision Letter · Decision Letter 2]

28 Jul 2021

Knowledge on Infection Prevention and Control and Associated Factors among Undergraduate Health Professional Students at Makerere University College of Health Sciences, Uganda

PONE-D-21-09130R2

Dear Dr. Nkalubo,

We’re pleased to inform you that your manuscript has been judged scientifically suitable for publication and will be formally accepted for publication once it meets all outstanding technical requirements.

Kind regards,

Prasenjit Mitra, MD, MRSB, MIScT, FLS, FACSc, FAACC

Academic Editor

PLOS ONE

Additional Editor Comments (optional):

Reviewers' comments:

Reviewer's Responses to Questions

**Comments to the Author**

1. If the authors have adequately addressed your comments raised in a previous round of review and you feel that this manuscript is now acceptable for publication, you may indicate that here to bypass the “Comments to the Author” section, enter your conflict of interest statement in the “Confidential to Editor” section, and submit your "Accept" recommendation.

Reviewer #1: All comments have been addressed

2. Is the manuscript technically sound, and do the data support the conclusions?

Reviewer #1: Yes

3. Has the statistical analysis been performed appropriately and rigorously? 

Reviewer #1: (No Response)

4. Have the authors made all data underlying the findings in their manuscript fully available?

Reviewer #1: Yes

5. Is the manuscript presented in an intelligible fashion and written in standard English?

Reviewer #1: Yes

6. Review Comments to the Author

Reviewer #1: The manuscript has been well written and all comments have been addressed. A few typographical and technical errors have been noted in the attached document.

7. PLOS authors have the option to publish the peer review history of their article (what does this mean?). If published, this will include your full peer review and any attached files.

Reviewer #1: No

---

## [Editor Report · Acceptance letter]

2 Aug 2021

PONE-D-21-09130R2 

Knowledge on Infection Prevention and Control and Associated Factors among Undergraduate Health Professional Students at Makerere University College of Health Sciences, Uganda 

Dear Dr. Nkalubo:

I'm pleased to inform you that your manuscript has been deemed suitable for publication in PLOS ONE. Congratulations! Your manuscript is now with our production department. 

Kind regards, 

on behalf of

Dr. Prasenjit Mitra 

Academic Editor

PLOS ONE